# A Posture Training System Based on Therblig Analysis and YOLO Model—Taking Erhu Bowing as an Example

**DOI:** 10.3390/s25030674

**Published:** 2025-01-23

**Authors:** Bonnie Lu, Chao-Li Meng, Chyi-Ren Dow

**Affiliations:** Department of Information Engineering and Computer Science, Feng Chia University, Taichung 40724, Taiwan; crdow@fcu.edu.tw

**Keywords:** erhu bowing, Therblig analysis, posture training, YOLO, object detection

## Abstract

Computer-assisted learning can help erhu learners to analyze their playing performance and identify areas for improvement. Therefore, in this study, a computerized system based on a You Only Look Once (YOLO)-OD model was developed for erhu bowing training. First, Therblig analysis was performed to segment the erhu bowing process into multiple steps, which were then analyzed thoroughly to identify critical objects for detection. Second, a YOLO-OD model was developed to detect and track the critical objects identified in video frames. Third, scoring methodologies were developed for bow level and bow straightness. The YOLO-OD model and the aforementioned scoring methodologies were incorporated into a computerized training system for erhu bowing, which enables erhu learners to practice independently. It provides scores for bow level and bow straightness, allowing learners to evaluate their technique, as well as feedback and instant alerts regarding incorrect motions and postures, which enable learners to adjust their actions and postures in real time. In addition, teachers or coaches can refer to the videos and other data collected using the proposed system in order to identify problematic erhu bowing techniques and provide students with relevant suggestions and feedback.

## 1. Introduction

Hands-on practice is essential when people learn to use tools or instruments [1,2]. Learners are typically required to interpret knowledge on their own when using traditional textbooks or video demonstrations. Although teachers can provide face-to-face guidance, it is difficult to measure whether the students have achieved their learning goals during self-practice. To facilitate self-practice at any time, an increasing number of self-learning systems incorporating sensing technologies have been developed worldwide [3,4,5,6,7]. These systems are designed to complement traditional learning methods, which often rely solely on text or video input. Some researchers have attempted to use sensor systems for learning [8,9,10,11,12,13,14]; for example, webcams can be used to capture and record dynamic images of learners in their learning environment for instant computer analysis. Such analysis can provide real-time feedback and performance information to learners, helping them to more effectively develop their skills. Training is essential in industrial contexts, helping to improve the quality and efficiency of the work carried out by employees. Consequently, augmented reality and virtual reality systems have been developed for training and maintenance [15,16]; however, these systems are more complex and expensive than video capture systems with object detection and tracking functions. Instrument operators must adhere to the relevant standard operating procedures, which can be monitored using computer vision technology; for example, this technology has been adopted to detect the posture of the operator and the positions of critical objects in recorded videos, allowing for analysis of the relationships between them to determine whether the relevant actions have been performed correctly.

Convolutional neural networks (CNNs) are commonly used to detect and classify objects in images. A video contains a sequence of image frames; therefore, methods used to track objects in images can also be used to track objects in a video. Redmon et al. proposed the You Only Look Once (YOLO) model for object detection [17]. YOLO is a one-stage object detection model in which a target image is processed only once using a pre-trained CNN model to obtain the output; thus, this model has an extremely high detection speed. The YOLO output comprises prediction results that include the location of the bounding box, class names, and confidence probabilities for the objects detected in the image. Improved versions of the YOLO model have been proposed; for example, Redmon et al. proposed YOLOv3 in 2018 [18], Bochkovskiy et al. proposed YOLOv4 and YOLOv7 [19,20,21], and Ultralytics developed YOLOv5, YOLOv8, and YOLOv11 [22,23]. Of these models, YOLOv5, YOLOv8, and YOLOv11 are based on the PyTorch framework, while YOLO, YOLOv3, YOLOv4, and YOLOv7 are based on the Darknet framework. In addition, many other YOLO models have been proposed by other researchers [24,25]. In the present study, we use YOLOv8, which achieved high training and inference speeds in the analysis of erhu playing techniques.

The erhu is a well-known traditional Chinese stringed musical instrument [26]. In this study, a YOLOv8 model was used to detect the positions of an erhu player’s right hand and their erhu bow in a sequence of video images, in order to analyze the bow proficiency of the player. Erhu teachers focus on teaching beginners how to steadily straighten the erhu bow. Straightening the bow is a crucial part of an erhu performance. Most beginners find it difficult to maintain the level and straightness of the bow during bow movements; these factors directly affect the timbre and consistency of the note produced by the erhu [27]. For novice erhu players who aim to perfect their bowing technique, it is essential to engage in persistent practice to cultivate proper habits. Although instructors can correct erroneous movements by providing face-to-face guidance or remote video conferences, their availability for continuous support may be limited. Through the use of computer-assisted training systems that are capable of tracking movements and evaluating the conformity to certain standards, learners gain the flexibility to practice at their convenience. In this study, we identified the bow level and bow straightness requirements for erhu bowing [28], and then designed methods for scoring the bow level and bow straightness. Subsequently, we developed a training system which helps erhu learners to keep their bow level and straight.

When designing a training system related to motion recognition, it is necessary to decompose the actions to thoroughly explore the sequence of movements and the relevant objects that need to be detected. Industrial engineering provides various methods for motion decomposition, including visual motion analysis, operation procedure analysis, Therblig analysis, film analysis, and image analysis. Among these, Therblig analysis is most commonly used to break down actions into basic elements, where complex body movements can be viewed as combinations of various motion elements. Therblig analysis also highlights the objects manipulated by the operator in each action. Therefore, when identifying images depicting mechanical motion, conducting a Therblig analysis beforehand to break down the elements of movement and identify related components will aid in the systematic identification and assessment of this mechanical action.

Therefore, we propose a framework including Therblig analysis and a YOLO model to analyze the actions of novice erhu learners in videos, which can be effectively included in posture training systems. The proposed framework can be extended to other fields, such as medical care and healthcare, where patients must use certain drugs or devices (e.g., an asthma inhaler) correctly, according to the relevant instructions [29,30]. The remainder of this paper is organized as follows. Section 2 presents a review of the related literature. Section 3 provides an overview of the proposed framework. Section 4, Section 5 and Section 6 detail the three stages of the proposed framework; namely, (1) posture requirement analysis, (2) posture understanding, and (3) training system development. Section 7 presents a discussion of the results, the limitations of this research, and recommendations for future work. Finally, Section 8 presents the conclusions of this study.

## 2. Related Work

### 2.1. YOLO Models

Redmon et al. proposed the first YOLO model (YOLOv1) [17]. YOLOv1 contains 24 convolutional layers followed by two fully connected layers, and this model outputs the class, coordinates, and probability of each object detected in an image. Redmon et al. also proposed YOLO9000 [31] and YOLOv3 [18] as improved versions of YOLOv1. YOLOv3 has a base network with 53 layers; this network is called Darknet-53 and can detect up to 9000 classes of multi-scale objects. YOLOv3 and subsequent YOLO architectures generally contain three parts: a backbone, neck, and head.

**Backbone:** The backbone contains a combination of CNNs that extract useful features from an input image.**Neck:** The neck, which connects the backbone to the head, aggregates and refines features at different scales.**Head:** The head comprises the output prediction layers. In these layers, the number of classes can be modified according to the application and data set.

Bochkovskiy continued the development of Darknet and soon released YOLOv4, along with Chien-Yao Wang and Hong-Yuan Mark Liao [19,20]. YOLOv4 is an improved version of YOLOv3 which is optimized for speed and accuracy. Glenn Jocher—the Chief Executive Officer of Ultralytics—developed a PyTorch framework for YOLO and proposed YOLOv5 in the same year that Bochkovskii et al. proposed YOLOv4. This framework has a simplified user interface and setup of the environment, and can thus accelerate YOLO development. YOLOv6, YOLOv7, YOLOv8, and YOLOv11 are based on the PyTorch framework [21,23,32].

Existing YOLO models support multiple scales, multiple tasks, and transfer learning. High accuracy can be achieved rapidly with these models, using a suitable pre-trained model and its weights as pre-existing knowledge. YOLOv4 and subsequent YOLO models can perform image classification, object detection, and semantic segmentation. One of the tasks added since YOLOv7, called YOLO-pose, performs pose estimation to detect key points in an image, such as joints of the human body. Therefore, in this study, we use YOLOv8 to perform object detection and pose estimation to determine the correctness of the bowing motion performed by erhu players.

### 2.2. Therblig Analysis

Therbligs, the concept of which was proposed by Frank Bunker Gilbreth between 1912 and 1915, are the basic actions required to complete a job. Frank Bunker Gilbreth stated that any operation is completed using one or more of 17 basic actions, which are called Therblig elements. The American Society of Mechanical Engineers later added the “find” element to the aforementioned 17 Therblig elements [33].

Initially, Therblig analysis was used to find and eliminate ineffective activities in production. Later, researchers used Therbligs to analyze assembly operations and human–machine interactions, as well as in action recognition [34,35,36]. Table 1 lists the 18 Therblig elements and their abbreviations.

In this study, Therblig analysis was employed to identify the key objects and motions essential for bow posture assessment in erhu performance. Initially, we aimed to detect four objects: the bow-holding right hand, the bow, the erhu body, and the performer’s face. However, the lack of an object–action relationship analysis limited the accuracy of posture recognition. Many traditional motion analysis methods focus primarily on the timing and sequencing of movements, often overlooking the interactions between objects and actions.

In contrast, Therblig analysis provides a structured framework to capture these relationships, enabling us to pinpoint the bow-holding right hand and the bow as the critical objects for tracking with our YOLO-OD model. This simplification reduced the complexity of the detection framework, thus improving the accuracy and efficiency of the system. In addition, YOLO-pose was adapted to determine the bowing center point, eliminating the need to track the erhu body and the performer’s face. Compared to other methods, Therblig analysis stands out for its ability to go beyond time-focused or process-level analysis through incorporating detailed motion categorization and object–action relationships. This capability makes it particularly suitable for tasks such as bow posture assessment in erhu performance, where understanding the interactions between objects (e.g., the bow and the hand) and key points of the human body is critical.

### 2.3. Computer-Assisted Learning for Musical Instruments

One can learn to play a musical instrument from a teacher or through self-study material. In recent years, Internet of Things (IoT) technology has been used for piano, guitar, drum, violin, and erhu instruction [4,6,7,8,9].

The authors of [37] created a violin bow with a carbon-fiber sensing system that measures the position and acceleration of the bow. However, little research has focused on computer-assisted systems for learning to play the erhu. A previous study [8] on erhu bowing used magnetic position sensors to display bow strokes in a virtual three-dimensional space, where different colors were used to provide feedback to learners in real time.

Bowing techniques for the erhu require the player to control the bow so that it is level and straight during its movement using their right hand. In general, a teacher visually evaluates whether the player is successful in achieving such control. Some studies have used cameras to detect bow posture [10,12,38,39]; however, an extremely limited number of studies (e.g., [38,39]) have used cameras to track bow movements.

A study on violin training from 2008 [40] highlighted the usefulness of integrating auditory alerts with visual feedback and score-tracking algorithms to assist trainees. Similarly, recent advances in computer-assisted systems have emphasized real-time scoring and visual feedback as effective tools to improve the understanding of erhu learners regarding their performance [41]. These systems dynamically evaluate performance metrics, allowing immediate insights and actionable feedback, which are crucial for improving technique during practice sessions but are generally too complex for users.

## 3. Overview of the Proposed System

The correctness of the actions performed during a task can be identified based on expert guidance, on-site observations, video analysis, or detected data. These actions can be analyzed to identify a sequence of actions for future improvement. In this study, a three-stage framework based on a YOLO model is proposed to develop a pose training system (Figure 1). The three stages of this framework are (1) posture requirement analysis, (2) posture understanding, and (3) training system development. Each of these stages is described in the following text.

In the posture requirement analysis stage, the required actions are analyzed and divided into steps, the critical objects are selected, and the operator’s actions are associated with relevant critical objects. There exist many methods for analyzing and segmenting motions. In this study, we used Therblig analysis to examine the actions involved in erhu bowing. Under normal circumstances, when an individual plays the erhu, their left hand holds the body of the erhu, their right hand holds the head of the erhu bow, and they move the bow in the left and right directions (Figure 2). As the bow hair is sandwiched between the two strings of the erhu, the bow does not leave the body of the erhu when the instrument is played.

After analyzing the required posture, the objects, human posture, and human motions to be detected and tracked were identified. Subsequently, in the posture determination stage, YOLO models were trained to detect the aforementioned aspects of erhu playing. We used approximately 2000 images to train different YOLO models—YOLOv3, YOLOv4, YOLOv5, YOLOv6, YOLOv7, and YOLOv8—and compared their performance in detecting the required objects. Furthermore, a Python program was developed, which uses the PyTorch framework of YOLOv8 to analyze the motion of the erhu player and the movements of the bow.

In the development stage of the training system, a posture training system was developed for learners. The system developed in this study provides motion and posture correctness scores, feedback on incorrect motions and postures, and instant alerts for users to adjust their movements. Erhu players can adjust their actions and postures in real time according to the feedback and alerts provided. We developed scoring methodologies which quantify the level and straightness of the player’s bow and incorporated these methodologies into the developed system, as described in detail in Section 6.

## 4. Posture Requirement Analysis

The erhu bowing process was segmented into 10 steps through Therblig analysis (Figure 3). These 10 steps were then analyzed to determine the key training requirements.

### 4.1. Steps Involved in Erhu Bowing

The 10 steps identified through the Therblig analysis are described as follows:The erhu player must select and pick up an erhu.Although the erhu can be played in standing position, erhu learners are recommended to always sit on a chair when playing the instrument. In addition, the bottom of the erhu should lie on top of the left leg of the player (Figure 2).After the player is seated, they must hold the stem of the erhu with their left hand and then grasp the bow head with their right hand.The right hand, which is used to grasp the bow head, is also used to place the bow hair flat on top of the erhu barrel.The bow is then pulled across the strings from left to right.When the bow head reaches the right boundary or the player wishes to return to the left side, they move the bow from right to left (bow pushing). The cycle of bow pulling and bow pushing is called erhu bowing.The player can stop the bow movement at any time to end the practice.After the end of the practice, the player must fold the bow and the erhu body together.The player must then stand up and go to a suitable location to place the erhu.Finally, the player places the erhu at the suitable location identified.

### 4.2. Selection of Critical Objects

Our focus area was the right-hand skills required for erhu bow movements; thus, steps 5 and 6 are essential in this study. Therefore, the developed model must be able to detect the bow position in video frames to recognize changes in this position between frames. Furthermore, in order to judge whether the bowing movement is level or straight, the bow posture must be identified by connecting the positions of the bow head and bow tail. The bow head may be too small to detect, but is always located in the right palm of the erhu player. Thus, the combination of the bow head and the player’s right palm can be considered a key object to detect. We selected two critical objects: the erhu bow and the player’s right palm, which grasps the bow head. Figure 4 illustrates the process of recognition for the posture of the erhu bow.

### 4.3. Associations Between the Erhu Player and Critical Objects

When the two selected critical objects and their left–right movement loops are correctly identified, it indicates that the erhu player is performing steps 5 and 6 of the erhu bowing process correctly. In addition to examining whether the erhu player performs the aforementioned steps appropriately, an online erhu training system must be able to evaluate whether the player has the requisite right-hand skill; that is, whether the bow is sufficiently level or straight. In this study, we used a YOLO model to track the two critical objects in video frames and developed algorithms to measure the correctness of the player’s posture when they moved the erhu bow.

To determine which bow section touches the string, the erhu body—as the center of the bowing route—must be detected in the video images. If the center of the bowing route is determined, the width of the bow can be used to determine which bow section the player is using. The bottom of the erhu is assumed to lie on the left leg of the player (step 2); therefore, instead of selecting the erhu as the third critical object to be detected, we use the YOLO-pose model to find the center point between the player’s left eye and left shoulder to approximate the position of the erhu.

## 5. Posture Understanding

After identifying the posture requirements for erhu bowing, we trained a YOLO-OD model through transfer learning and used this model to detect the positions of the erhu bow and the player’s right hand in video frames. Subsequently, we used our proposed methods to track bow movements to determine the bow posture of the player and provide a correctness score for this posture based on the posture requirements.

Existing algorithms for analyzing erhu bowing techniques, such as VICON-based systems [40] or hybrid models (e.g., GCN+TCN) [39], primarily focus on high-precision motion analysis. These methods often rely on advanced equipment, such as optical motion capture systems or complex mathematical models, which provide accurate spatial and temporal analysis of bowing techniques. However, they are resource-intensive, costly, and less practical for real-time feedback in everyday learning environments.

The proposed system leverages the YOLO-Therblig framework to detect and track critical objects, including the erhu bow and the player’s right hand. It provides real-time feedback by assessing bow levelness and straightness, using simple yet effective scoring mechanisms. The system is designed for low-cost deployment, allowing broader accessibility and real-time corrections during practice sessions. Table 2 is a comparison of our proposed YOLO-based approach and these existing algorithms.

### 5.1. Development of the YOLO-OD Model

The images used to train the developed YOLO-OD model were captured from several Chinese music orchestras and societies in Taiwan. We extracted images from videos of 29 erhu players, only selecting images with a front view of the erhu player, in order to form the training set. Whenever someone is playing the erhu, their state of motion changes every few frames; therefore, we developed a script to automatically capture a screenshot of the videos every 2 s, thus obtaining a large amount of training data.

Before training the YOLO-OD model, we pre-processed the image data to address potential variability caused by environmental factors and camera configurations. For example, we manually deleted similar images and images that did not display the erhu player from the front view. To increase data diversity and account for environmental variability, we collected images from as many different conditions as possible, including indoor and outdoor environments, daytime and nighttime settings, and scenes with artificial lighting. Additionally, to improve robustness against variations in camera angles and distances, we included footage captured at different distances and tilt angles (e.g., high, low, and oblique).

Once the data collection and pre-processing phases were completed, we used an image annotation application that is compatible with the YOLO format to label the entire data set. YOLO models compress images into squares for training and detection, but the data set collected in this study included portrait and landscape images, as most cameras do not capture square images. To mitigate the distortions caused by resizing, we employed a padding method to pre-process all images into square dimensions, ensuring consistency between bounding boxes and object sizes during training. In cases where only portrait (or landscape) images were available for a certain player, we applied padding to create the corresponding landscape (or portrait) versions. This data augmentation method not only improved the robustness of the YOLO-OD model under varying conditions, but also enhanced the accuracy and intersection over union (IoU) score, as has been shown in our previous research.

To further refine the model’s performance, we iteratively expanded the data set whenever the accuracy was insufficient or certain types of images were under-represented. Additional suitable images were obtained from the Internet and incorporated into the training set, followed by subsequent rounds of training. These pre-processing and augmentation strategies significantly contributed to the model’s ability to perform consistently across diverse environmental conditions and camera configurations.

The collected data set contained approximately 2000 images, and was divided into three parts, with 70% (1524 images) used as the training data set, 20% (435 images) as the validation data set, and 10% (210 images) as the test data set. As the collected data might be insufficient to produce an accurate model, we used a YOLO model that had been pre-trained on large data sets to perform transfer learning in our YOLO-OD model. We trained our YOLO-OD model with YOLOv3 and achieved an accuracy greater than 95% on the test data set. Subsequently, we also trained our YOLO-OD model with other YOLO models, in order to compare their performance. Figure 5 displays the results of the YOLOv8n (the smallest YOLOv8 model) training for 100 epochs.

Table 3 lists the performances of the YOLO-OD models on the test set when they were trained using the YOLOv3-spp, YOLOv4, YOLOv5m6, YOLOv6m, YOLOv7, YOLOv8n, and RTDETR models. The accuracy, speed, and average IoU score of the YOLO-OD models were acceptable and improved with improvements in the YOLO model used for the testing. The average IoU of the YOLO-OD model was >13% higher when it was trained using YOLOv8n than when trained with YOLOv3. The test results for these object detection models on the test data set indicated a recall rate and an F1 score of nearly 100%. This indicates that any generation of YOLO, when trained on the data set we collected, can effectively detect the two critical objects—the bow and the bow-holding right hand—in most images. Among these models, we opted for the lighter one, YOLOv8n, due to its efficiency.

RTDETR is a recently proposed object detection model that distinguishes itself from YOLO models, which employ a fully connected neural network as the classifier in their output layer. RTDETR, on the other hand, replaces this classifier with a transformer neural network. This model not only utilizes a deep CNN for dimensionality reduction and feature extraction, but also omits the intermediate layer, training the encoder and decoder to extract key features and act as a classifier. Although it has been claimed that RTDETR outperforms YOLO on the COCO data set, the results with RTDETR in this study did not exceed those of YOLOv8. Compared to RTDETR, which requires multiple GPUs for training, YOLO models require fewer computational resources and thus are more cost-effective.

To verify the generalization performance of the YOLO-OD model, we tested it on 10 frames randomly sampled from each of the videos of 10 bow players. Of the 10 videos, videos V1–V7 were captured in two Chinese orchestras, while videos V8–V10 were obtained from YouTube. The model performance verification results are presented in Table 4.

The detection rate for V1 was lower than that for the other videos (Table 3), as V1 was shot close to the front. To increase the detection rate for V1, 10 additional images were randomly selected from V1 and added to the data set. After re-training the YOLO-OD model using these additional data, its detection rate for V1 improved considerably; however, its detection rates for the other videos were marginally affected, especially the detection rate for V9, which decreased to below 90%. Therefore, several additional images from V9 were also added to the data set and training was performed again. The results obtained after re-training with these images are presented in Table 5.

When the YOLO-OD model was adjusted and re-trained for any video, its detection rates for others were affected. Therefore, images of the two critical objects from the target players’ videos were added to the data set in order to dynamically improve and balance the performance of this model for all videos.

### 5.2. Detection Algorithms

After the YOLO-OD model was trained to detect the erhu bow and the player’s right hand, we used the YOLOv8 framework [23] to detect these two critical objects in video frames and record their x, y (i.e., width and height) coordinates for further analysis. If a camera is used, these analysis results can be displayed in real time. Figure 6 shows example bowing detection and analysis results.

To identify the bow section used by an erhu player, the center of the bowing path (which is located inside the erhu barrel) must be detected. Steps 2 and 3 of the bowing process indicate that the erhu barrel is placed above the left leg of the player and is held by the left hand of the player. Therefore, we performed pose estimation using YOLO-pose to extract key points from the left eye and left shoulder of the player. To avoid the detection of any additional objects—which leads to excessive computational complexity and compromises the overall accuracy of the model—the center of the line that connects the left eye of the player and the left shoulder was used to approximate the position of the erhu barrel. As illustrated in Figure 7, when the center point of the bow is on the right (left) side of the erhu, the right (left) half of the bow is in use. The bow section can be divided into three parts: the left, middle, and right bow sections.

### 5.3. Detection of Posture and Movement

An objective measure of the levelness of erhu bowing is the deviation from a horizontal line, as used and formulated by Kikukawa et al. [8] in their evaluation of the quality of bow posture based on magnetic position sensor data; however, they did not mention the straightness of the bow movement.

We developed relevant methods for assessing the bow level and bow straightness, as described in the following text.

#### 5.3.1. Bow Level

The bow’s hair should be level and close to the erhu barrel. When the bow movement is level, the right hand is very close to the horizon line in the middle of the bow’s bounding box, as illustrated in Figure 8a.

To determine the bow level, the YOLO-OD model must detect the erhu bow and the right hand of the erhu player. As the size of an object in an image depends on the distance between the camera and the object, using only the height of the bow’s bounding box to evaluate the bow level is impossible.

Therefore, we used the proportion of the height of the right hand as a tolerance parameter. When the center point of the right-hand bounding box is within the tolerance of the middle horizon line of the bow’s bounding box, the developed system considers the bow level-related criterion to be satisfied.

#### 5.3.2. Bow Straightness

Erhu players must perform straight bow movements during bowing. The slope of the bow can be quantified by a straightness indicator, as illustrated in Figure 8b. If the YOLO-OD model only detects the bounding box of the bow, the direction of bow movement cannot be determined. As the erhu player should always hold the bow head in their right hand, the center point of the right hand can be considered the start point of the bow. Therefore, to draw the bow line, the end point of the bow must be determined. If the start point of the bow is above (below) the middle horizon line of the bow’s bounding box, the end point of the bow is near the bottom-right (upper-right) point of this box.

The coordinate origin of the video frames is located in the upper left corner. To allow the learner to observe the inclination intuitively, the coordinate origin must be moved to the lower left corner before calculating the slope. Equation (Equation 1) is used to translate the coordinates of the start and end points of the bow from (x, y) to (x’, y’). Thus, when the bow is above the horizontal line of the bow’s bounding box, a negative slope is obtained; otherwise, a positive slope is obtained. This representation can confuse the learner; therefore, the slope is converted to its negative value, denoted as BSLP (bow slope), using Equation (Equation 2).(1)$$bowstart,bowend=\left(x_{s}^{\prime },y_{s}^{\prime }\right),\left(x_{e}^{\prime },y_{e}^{\prime }\right)=\left(x_{s},ImageHeight-y_{s}\right),\left(x_{e},ImageHeight-y_{e}\right),$$(2)$$BSLP_{eachFC}=-1\times \frac{\left(bowend\left[y_{e}^{\prime }\right]-bowstart\left[y_{s}^{\prime }\right]\right)}{\left(bowend\left[x_{e}^{\prime }\right]-bowstart\left[x_{s}^{\prime }\right]\right)}$$

## 6. Posture Training System

Erhu teachers are not always available to students when they face problems during self-practice. To understand the situations of the students, some teachers ask students to send practice videos, so that they can provide suggestions to the students. However, this is inconvenient and possibly unreliable. Moreover, erhu learners can only rely on their teacher’s memory or notes. Using the recording and reporting function of the proposed computer-assisted posture training system, erhu teachers can analyze the collected data, quickly identify students with problematic techniques, and send comments to these students. In addition, the erhu teachers can use the learning portfolio to formulate a suitable learning program for students.

### 6.1. System Design

As an erhu player would not always be watching the screen of the proposed system, the system provides an auditory warning if the player’s posture is not within the tolerance range. Tone A indicates that the bow head is above the target line, while tone D indicates that the bow head is below the target line. These tones are open-string sounds produced by an erhu, and the player can distinguish these tones clearly. Figure 9 shows the architecture of the proposed posture training system.

When an erhu player hears a sound warning from the proposed system, they know that they have committed a mistake in their movement and can then adjust their movement until the warning sound is eliminated. The proposed system can also output analysis reports as a reference for future improvement.

### 6.2. Scoring Methods

#### 6.2.1. Scoring Method for Bow Level

The scoring method for bow level is described as follows. Each video frame in which both critical objects are detected by the YOLO-OD model is considered a valid frame, while those that do not contain the critical objects or for which the YOLO-OD model cannot detect both critical objects are considered invalid frames. ’Total frames’ refers to the total number of video frames examined. If a valid frame satisfies the bow level criterion, the frame is considered a correct frame. The bow level score (BLS) (Equation 3) for a video is then defined as the ratio of the number of correct frames to the number of valid frames. An example of the BLS calculation is illustrated in Figure 6a.(3)$$BLS=\frac{\;Correct\;frames_{level}}{Total\;frames\;examined}.$$

When the user selects the ‘Bow Level’ mode, they must maintain the bow as horizontally as possible in each frame. The system can set the allowable range for upward and downward deviations. The proportion of detected right-hand objects is use to avoid the influence of the distance of the target objects in the image (e.g., 1/2, 1/4, or 1/6). The smaller the allowable error proportion, the higher the difficulty.

The BLS score is immediately displayed on the screen and can be recorded. The higher the score, the greater the number of bow image frames that meet the requirements. Learners can set improving their BLS scores as a learning goal, in order to maintain bow proficiency and ultimately achieve the desired erhu bowing technique. We also classify non-compliant bowing image frames as being above or below the allowable position range, such that learners can gain an in-depth understanding of their bowing posture problems.

#### 6.2.2. Scoring Method for Bow Straightness

The scoring method for bow straightness is described in the following text. As in the scoring method for the bow level, each video frame in which both critical objects are detected by the YOLO-OD model is considered a valid frame; meanwhile, frames that do not meet this criterion are considered invalid frames. A frame in which the BSLP line is located within the accepted straightness tolerance is considered a correct frame. The bow straightness score (BSS) (Equation 4) is calculated as the ratio of the number of correct frames to the number of valid frames. An example of the BSS calculation is illustrated in Figure 6b.(4)$$BSS=\frac{\;Correct\;frames_{Straightness}}{Total\;frames\;examined}.$$

When the user selects the “Bow Straightness” mode, they must strive to maintain a specific slope value of the bow in each frame, with the “Bow Level” mode equivalent to having a slope of 0. The system can set the target slope value that the learner intends to achieve and specify the acceptable range for both upward and downward deviations. Unlike the ‘Bow Level’ mode, which uses the proportion of the right-hand object as the relative error value, the ‘Straightness’ mode directly employs the slope value as the upper and lower bounds of permissible deviation. The lower the allowed deviation value, the higher the level of difficulty.

The BSS score is displayed on the screen and can be recorded. A higher score indicates a higher number of bowing image frames that meet the requirements. Learners can use improving their BSS scores as a learning objective in order to maintain bow straightness and ultimately achieve the desired erhu bowing technique. Additionally, we categorize non-compliant bowing image frames as either above or below the acceptable position range, providing learners with a thorough understanding of their bowing posture issues.

The distance between the upper and lower BSLP, denoted as the bow straightness bandwidth (BSB), can be considered as a quantity that describes the straightness capability of the player, defined as in (Equation 5).(5)$$BSB={\left(max\left(BSLP\right)-min\left(BSLP\right)\right)}_{Total\;frames\;examined}.$$

The BSB indicator measures the difference between the upper and lower limits of the learner’s bow slope across all bowing image frames in the video. A smaller BSB value indicates a narrower range of upper and lower deviations, reflecting the learner’s improved ability to control straightness. Minimizing the BSB value can be set as a learning goal to achieve the required level of straightness in erhu bowing.

Furthermore, this system provides a visual representation chart of the BSB, as shown in Figures 11 and 12, providing instructors and learners with the ability to analyze erhu bowing issues and propose appropriate improvement plans. The key information presented in the image includes the following:Trajectory of the slope of the bow during bow movement: We applied the algorithm discussed in Section 5.2 to determine the learner’s bow-pull and bow-push phases. In the image, green represents the bow pull and blue represents the bow push. The frame-by-frame depiction of the bow’s up-and-down movement in the video, along with the blue and green colors, illustrates changes in the bowing direction. These data not only help professionals to visualize changes in the learner’s bow movements, but also enable future computer data analysis to provide automated suggestions.Offset position of the BSB relative to the baseline: The upper or lower boundary range of the BSB indicates the learner’s bowing habits. This chart allows for the identification of shortcomings in the learner’s habits, facilitating appropriate adjustments.BSB overlap range between bow pull and bow push: We separately calculate the BSB for the bow pull and bow push. The larger the overlap range ratio between the upper and lower boundaries of these two phases, the more stable the learner’s bow-handling habits. We denote the overlap range ratio as ${\mathrm{IOU}}_{BSB}$. As higher BSBs may have higher overlap rates, another metric, ${\mathrm{IO(U)}}_{BSB}^{2}$, serves to normalize the difference based on the union of the BSB ranges at each time.(6)$$IOU_{BSB}=\left(BSB_{pull}\cap BSB_{push}\right)/\left(BSB_{pull}\cup BSB_{push}\right),$$(7)$$IO{\left(U\right)}_{BSB}^{2}=IOU_{BSB}/\left(BSB_{pull}\cup BSB_{push}\right).$$

### 6.3. Interactive Learning Outcomes with System Assistance

Crossover experiments were conducted to assess the impacts of the system on the learning improvements of the erhu learners during practice, comparing the performance of the participants with and without assistance from the system. Two groups were formed:**Group 1:** Initially used the system before discontinuing its use. Learners A, B, and C were in group 1.**Group 2:** Began without the system and later incorporated it into their practice. Learners D, E, and F were in group 2.

This experimental design facilitated a comparison of learners’ progress under both conditions, ensuring that all participants experienced both assisted and non-assisted practice phases.

To ensure consistency, all participants were novices with no previous experience learning erhu. To minimize fatigue, a two-minute rest was provided between each two-minute practice session. During system-assisted practice, learners received real-time feedback in the form of a warning sound for incorrect posture and visual displays of bow posture and bowing scores on the screen. In contrast, when the system was not used, no real-time feedback was given and only the learners’ performance was recorded for later analysis. At the end of each measurement period, the observer signaled to the participant to stop.

The erhu bowing technique involves a continuous left–right reciprocating motion of the hand, where each upward movement is followed by a downward return and vice versa, which can result in high values for both BSB (Equation 5) and ${\mathrm{IOU}}_{BSB}$ (Equation 6), as was the case for learners C and D. However, as the learner’s BSB narrows, the stability improvement may not be easily apparent from ${\mathrm{IOU}}_{BSB}$ measurements alone. To address this limitation, we introduced ${\mathrm{IO(U)}}_{BSB}^{2}$ (Equation 7), which is calculated by normalizing the IOU with union again. A higher ${\mathrm{IO(U)}}_{BSB}^{2}$ value indicates a narrower BSB within a high IOU, offering a more precise metric for assessment of the stability of the bow stroke.

Table 6 presents the BLS scores (Equation 3) and various BSB performance indicators for the six participants, comparing their results with and without the system. Figure 10 illustrates the average score trajectories for both groups, with the first attempt on the blue line and the second attempt on the green line. The learners who used the system consistently achieved higher BLS scores. Regarding BSB indicators, most learners (except for learner C, who obtained a lower ${\mathrm{IO(U)}}_{BSB}^{2}$ without the system) showed improved BSB performance after using the system. This indicates that the system effectively influences the learners’ practice behaviors, helping them to meet the level and straightness requirements for proper erhu bowing.

One person was chosen from each group to graphically illustrate their slope value trajectories across continuous frames in their two attempts, along with their respective BSB indicators, as shown in Figure 11 and Figure 12.

These two participants both exhibited high BSB values when not using the system, indicating areas for improvement. Upon examination of those video frames with higher slope values, several teachers identified that the arms of these two learners were excessively extended during bowing. This excessive opening angle causes the hand to lift the bow—a common issue among novice erhu learners. To address this, the hand axis should push toward the front when the string touches the left bow section to maintain horizontal alignment and bow straightness. Another issue observed for learner D was that he did not move his hand axis backward while pushing the bow; instead, he only quickly moved his upper arm to his body. This caused the bow head to sink under the horizontal alignment.

Through the analysis of various indicators, incorrect postures can be identified, allowing for computer-assisted analysis to provide improvement suggestions in real time, significantly reducing the time teachers must spend reviewing videos to detect problems.

## 7. Discussion

In this study, Therblig analysis was carried out to break down the erhu bowing process into sequential steps. These steps were examined to identify critical objects and their relationships with key points of the human body. A YOLO-OD model was trained and employed to track these critical objects, with the positions of key points of the erhu player’s body determined using the official YOLOv8n pose model. Our proposed posture and movement detection algorithms were utilized to assess whether the actions performed met the required level and straightness criteria. The proposed erhu training system, with real-time scoring and warning functions, empowers erhu learners to practice independently and self-evaluate their technique against established standards, without an instructor on site. In addition, instructors can use the data collected by the proposed system to identify learners with erhu bowing technique issues and provide targeted suggestions and feedback, rather than spending extensive time reviewing the learners’ videos.

Although the proposed training system is beneficial for enhancing bowing technique in erhu learners, it possesses certain limitations that require further investigation in future studies:If the performer in the video is not directly facing the camera or if the shooting angle differs from the images in the YOLO-OD model training data set, the proposed system may yield errors in assessing bow posture. Similarly, variations in the camera’s tilt angle (e.g., high or low angles) can distort the spatial relationship between key points, further affecting the accuracy of posture assessment. To address these issues, posture key points can be utilized to calculate the body’s angles, allowing for deduction of the bow’s posture as if the body were oriented towards the front. Future iterations of the model could incorporate methods for correcting or normalizing data captured from tilted perspectives. Additionally, the performance of the model may be influenced by variations in environmental conditions; for example, differences between indoor and outdoor settings, changes in natural or artificial lighting, and background complexity could introduce inconsistencies in data quality. These factors may reduce the robustness of the model when applied in diverse scenarios. Expanding the training data set to include diverse camera angles and environmental conditions, as well as exploring normalization techniques, could enhance the system’s adaptability and reliability.Maintaining a relaxed right wrist while holding the bow is another fundamental requirement. When drawing the bow, the player’s wrist bends inward, while when pushing the bow, it extends outward. However, it is challenging to discern the wrist flexion from a frontal view as the palm obscures the wrist. This issue can be resolved by employing IoT devices, such as accelerometers, or by training the YOLO-pose model to identify the right hand’s key points. Alternatively, using the YOLO segmentation model to analyze the proportion of the inner and outer areas of the right palm can help in determining the wrist’s flexion state.

In addition, we recommend incorporating the following functions into the training system in future research:The section of the bow being used, along with the angle of the player’s upper right limb relative to the body, can provide insights into whether the right arm is moving into the correct position. For example, the angle between the performer’s right arm and the body should not exceed 45 degrees when using the right half of the bow and 30 degrees when using the left half of the bow.Another prevalent issue for beginners of the erhu is that the bow hair occasionally leaves the sound barrel. Some studies have used magnetometers to detect this occurrence. Our YOLO models, in conjunction with our bowing detection algorithm, can calculate the movement of the intersection point between the bowing center line and the bow posture line in the vertical direction. This analysis might help in determining whether the bow is being lifted.Through detecting the sections of the bow that are frequently used by the player and calculating the length of bow utilized, the system can remind the erhu player to maximize their use of the full bow. It can also determine the player’s technique by calculating the length of each bow movement and converting the frame speed. This allows for the identification of various bowing methods, such as long bow, short bow, jumping bow, and throwing bow.

Researchers can leverage the integration of Therblig analysis and YOLO models to develop training systems tailored to specific requirements, while also analyzing relevant recorded data to enhance the efficiency of teaching. The proposed system framework demonstrates flexibility and adaptability, making it applicable in various domains. Potential applications include training in the use of medical equipment, physical activities that involve sports equipment, warehouse operations, assembly manufacturing, and tool manipulation. Furthermore, this framework could be extended to posture training tasks, such as musical instrument performance or rehabilitation exercises, where understanding object–action interactions is critical.

## 8. Conclusions

Traditional teaching methods often face challenges in terms of systematically recording and analyzing learners’ practices. Computer-assisted learning, based on sensing technology and computer vision, offers a promising solution that helps learners evaluate their performance and identify areas for improvement. This study developed a computerized system for learners of erhu bowing. Through Therblig analysis, the bowing process was segmented into multiple steps, enabling the identification of critical objects for detection. YOLO-based models were then developed to detect human and bow movements, as well as their postures. Furthermore, methods were proposed to calculate the BLS, BSS, BSB, and ${\mathrm{IO(U)}}_{BSB}^{2}$ metrics, providing a convenient and quantitative evaluation of the erhu bowing techniques of the learners; specifically, to evaluate the level and straightness requirements.

Although the proposed system achieved acceptable accuracy under various conditions, a limitation lies in assessing subtle movements, such as wrist flexion and extension. The frontal view used in this study obscures wrist motion due to the position of the palm. To address this challenge, future enhancements could include integrating IoT devices, such as accelerometers, to capture wrist movement data, training the YOLO-pose model to detect additional key points, or employing YOLO classification and segmentation models for combinatorial judgment.

In addition, future research should focus on establishing operational benchmarks for various bowing techniques, providing a more standardized framework for evaluation. We also encourage extended studies by other researchers to further enhance the means of practical assistance for the erhu community. These potential improvements underline the flexibility and scalability of the proposed framework, making it applicable not only to erhu training but also to other posture training tasks in music, sports, and beyond.

## Figures and Tables

**Figure 1 sensors-25-00674-f001:**
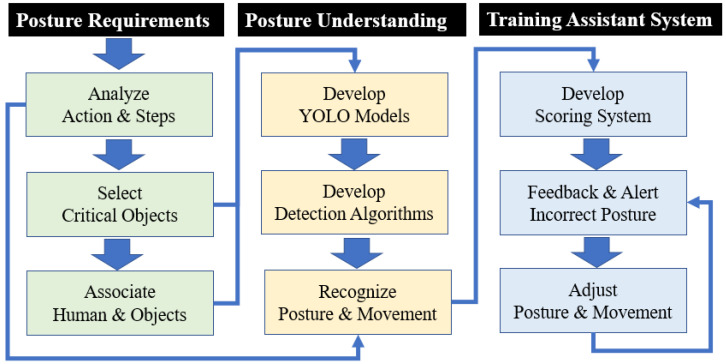
Posture training system overview.

**Figure 2 sensors-25-00674-f002:**
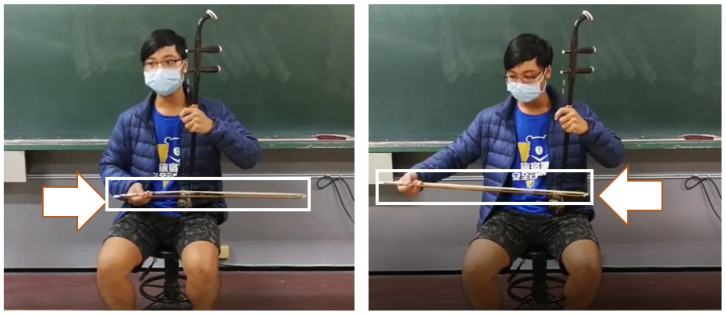
An example of a student of Feng Chia University playing the erhu.

**Figure 3 sensors-25-00674-f003:**
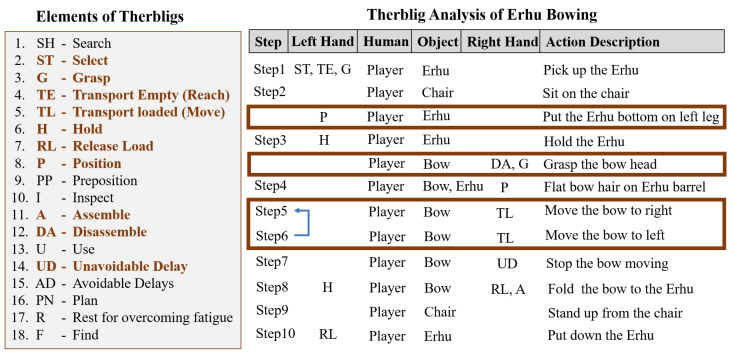
Therblig analysis of erhu bowing operation.

**Figure 4 sensors-25-00674-f004:**
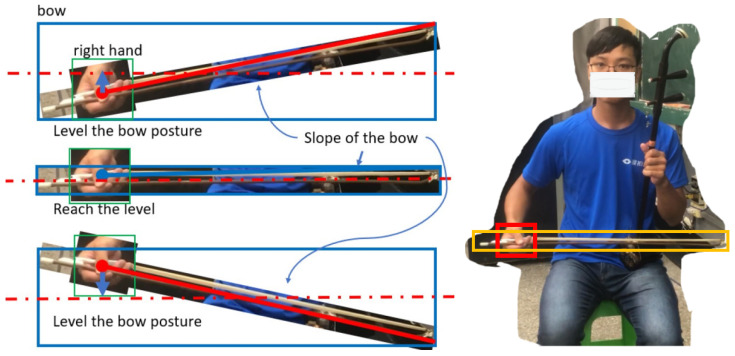
Process for recognizing the posture of the erhu bow during bowing movement.

**Figure 5 sensors-25-00674-f005:**
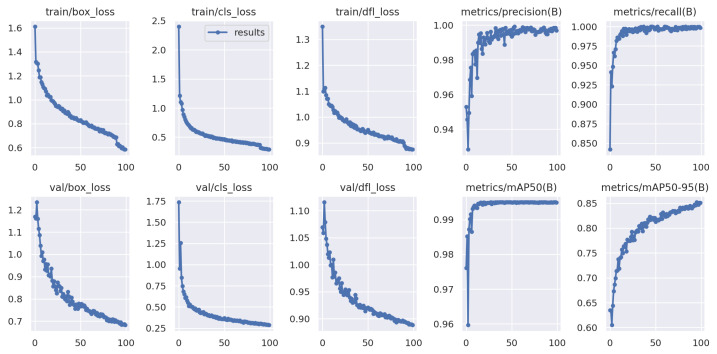
The training results of our YOLO-OD model using YOLOv8-n.

**Figure 6 sensors-25-00674-f006:**
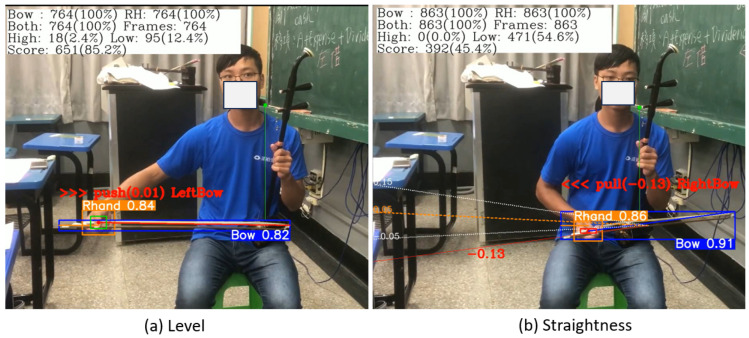
Examples of the bowing detection results and analysis outcomes.

**Figure 7 sensors-25-00674-f007:**
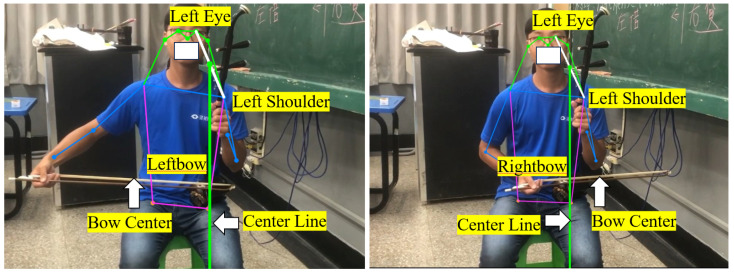
Detection of bowing center point using YOLO-pose.

**Figure 8 sensors-25-00674-f008:**
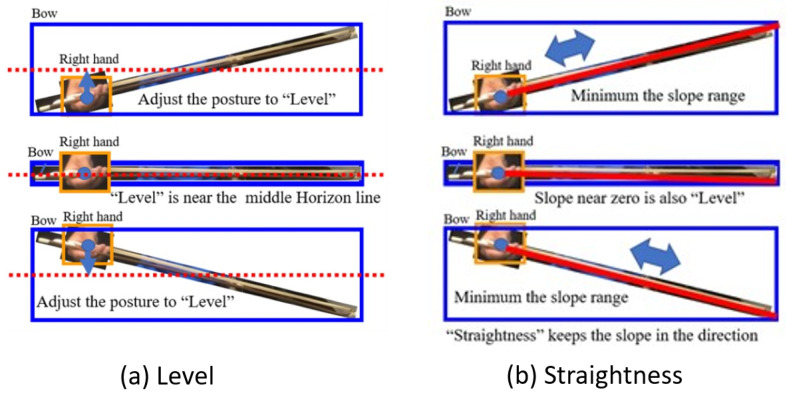
The concept of correct level and straightness for erhu bowing.

**Figure 9 sensors-25-00674-f009:**
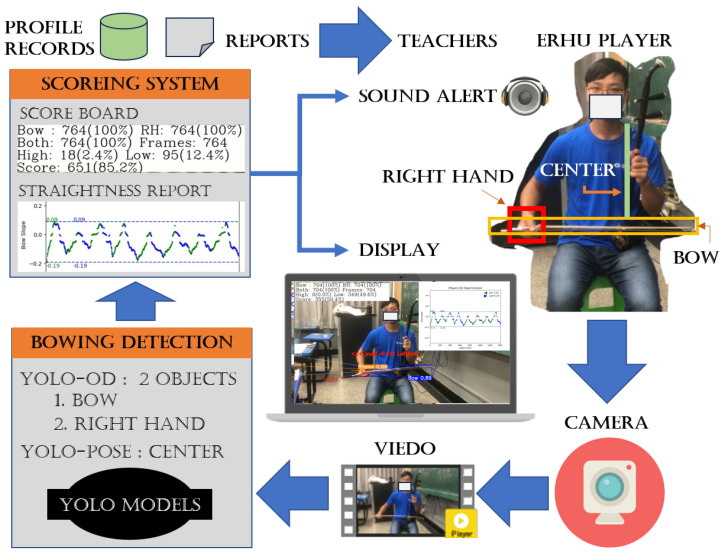
Architecture of our proposed posture training assistant system.

**Figure 10 sensors-25-00674-f010:**
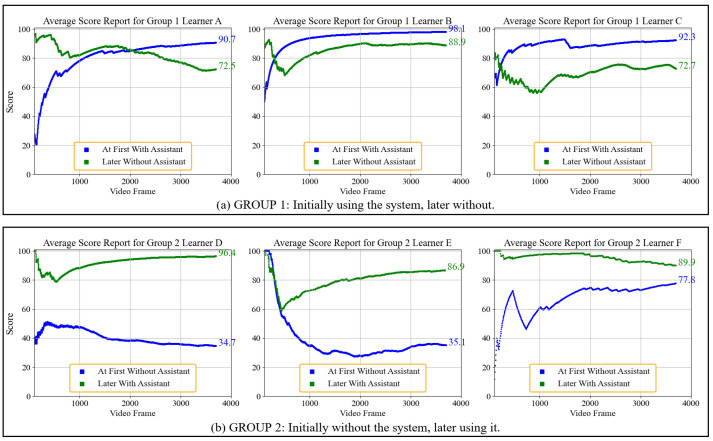
BLS score analysis: Erhu bowing performance in experimental groups.

**Figure 11 sensors-25-00674-f011:**
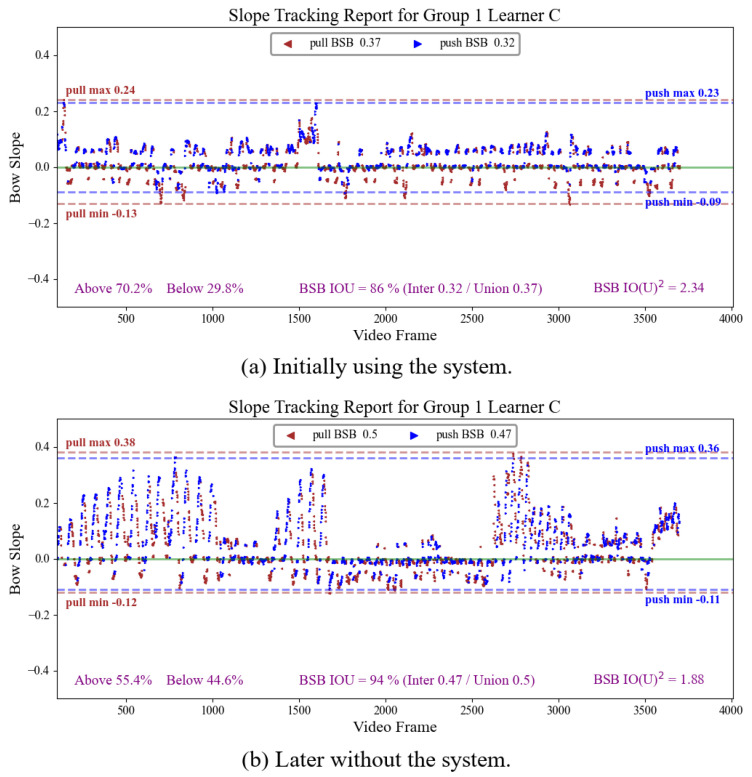
Slope tracking report of erhu bowing for learner C.

**Figure 12 sensors-25-00674-f012:**
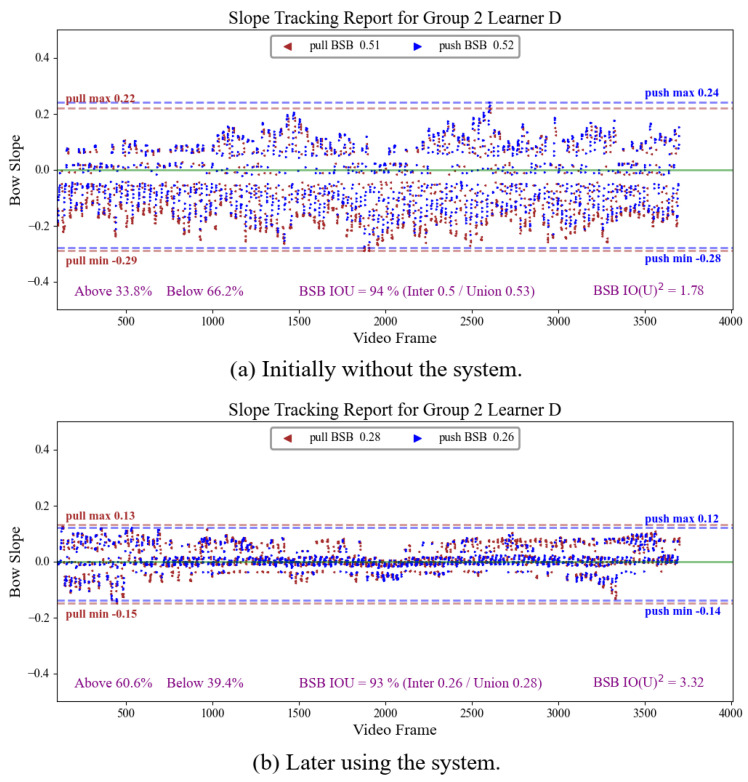
Slope tracking report of erhu bowing for learner D.

**Table 1 sensors-25-00674-t001:** Elements used in Therblig analysis.

Therblig	Code	Description
1. Search	SH	Seeking an object using the eyes and hands
2. Select	ST	Choosing among several objects
3. Grasp	G	Grasping an object with a hand
4. Transport Empty (Reach)	TE	Receiving an object with a hand
5. Transport Loaded (Move)	TL	Moving an object using a hand motion
6. Hold	H	Holding an object
7. Release load	RL	Releasing control of an object
8. Position	P	Positioning or orienting an object in the defined location
9. Preposition	PP	Positioning or orienting an object for the next operation and relative to an approximation location
10. Inspect	I	Determining the quality or the characteristics of an object using the eyes and/or other senses
11. Assemble	A	Joining multiple components together
12. Disassemble	DA	Separating multiple components that were joined
13. Use	U	Manipulating a tool in the intended way during working
14. Unavoidable Delay	UD	Waiting due to factors beyond the worker’s control
15. Avoidable Delay	AD	Waiting within the worker’s control that causes idleness
16. Plan	PN	Deciding on a course of action
17. Rest	R	Resting to overcome fatigue (e.g., pausing during a motion)
18. Find	F	A momentary mental reaction at the end of the search

**Table 2 sensors-25-00674-t002:** Comparison of existing algorithms and proposed YOLO-based approach.

Feature	Existing Algorithm	Proposed YOLO-Based Approach
**Detection Method**	Optical motion capture (e.g., VICON) or GCN+TCN models	YOLO models for object detection and pose estimation
**Cost**	High (specialized hardware required)	Low (requires only a camera)
**Real-Time Feedback**	Limited or post-analysis	Immediate feedback with sound and visual alerts
**Scalability**	Limited (requires specific setups)	High (can be deployed in home or classroom environments)
**Precision**	Extremely high, for professional use	Sufficient for educational purposes
**Ease of Use**	Complex setup and high computational requirements	Simple deployment and low computational overhead
**Learning Progress Analysis**	Detailed error analysis and progress tracking	Quantitative metrics (e.g., BLS, BSS) for tracking progress
**Target Users**	Professionals and researchers	Beginners and general learners

**Table 3 sensors-25-00674-t003:** The performance outcomes of different YOLO-OD models.

Performance	YOLO	YOLO	YOLO	YOLO	YOLO	YOLO	RTDETR
	v3-spp	v4	v5m6	v6m	v7	v8n	Large [42]
mAP—Bow	95.3%	99.5%	100.0%	99.9%	99.5%	100.0%	99.6%
mAP—Right Hand	99.5%	100.0%	100.0%	99.9%	99.5%	99.9%	99.6%
mAP@0.50	97.4%	99.8%	99.5%	99.5%	100.0%	99.5%	99.5%
avgIoU	76.3%	84.5%	90.1%	88.8%	90.5%	90.1%	89.3%
F1 score	0.96	1.00	1.00	1.00	1.00	1.00	1.00
Recall	0.94	1.00	1.00	1.00	1.00	1.00	1.00
Layers	114	162	323	194	314	168	498
Params (M)	134.2	134.2	41.1	51.9	36.4	3.0	31.9
FLOPs (G)	140.3	127.2	65.1	161.1	103.2	8.1	110

**Table 4 sensors-25-00674-t004:** The performance outcomes of our YOLO-OD model for different videos.

Performance	V1	V2	V3	V4	V5	V6	V7	V8	V9	V10
mAP—Bow	98.9%	98.4%	100.0%	100.0%	99.3%	100.0%	91.9%	99.5%	90.0%	100.0%
mAP—Right Hand	88.4%	99.5%	100.0%	99.5%	98.4%	100.0%	99.4%	100.0%	100.0%	90.0%
mAP@0.50	98.2%	99.5%	99.5%	99.5%	99.5%	99.5%	99.5%	97.3%	96.0%	95.0%
avgIoU—Bow	81.5%	94.8%	85.8%	89.2%	86.1%	84.7%	94.5%	83.3%	74.3%	90.5%
avgIoU—RH	60.5%	74.6%	82.9%	72.1%	80.7%	85.8%	82.6%	82.7%	82.5%	71.8%
F1 score	0.83	1.00	1.00	1.00	1.00	0.99	0.98	0.97	0.95	0.95
Recall	0.95	1.00	1.00	1.00	1.00	0.99	1.00	0.95	0.95	0.95
Total frames	812	447	1941	1947	827	841	866	1462	3145	1499
Bow detected	806	447	1941	1930	798	756	866	1448	2987	1499
	(99%)	(100%)	(100%)	(99%)	(96%)	(89%)	(100%)	(99%)	(94%)	(100%)
RH detected	630	443	1940	1932	806	841	853	1434	3026	1498
	(77%)	(99%)	(99%)	(99%)	(97%)	(100%)	(98%)	(98%)	(96%)	(99%)
Both detected	626	443	1940	1915	777	756	853	1421	2893	1498
	(77%)	(99%)	(99%)	(98%)	(93%)	(89%)	(98%)	(97%)	(91%)	(99%)

**Table 5 sensors-25-00674-t005:** The performance outcomes of YOLO-OD model re-trained with images from V1 and V9.

Performance	V1	V2	V3	V4	V5	V6	V7	V8	V9	V10
mAP—Bow	98.5%	100.0%	98.5%	99.5%	97.9%	98.6%	92.6%	99.0%	99.3%	98.4%
mAP—Right Hand	99.6%	99.5%	100.0%	99.5%	90.6%	100.0%	100.0%	99.2%	98.4%	100.0%
mAP@0.50	99.5%	99.5%	99.5%	99.5%	98.6%	99.5%	99.5%	96.8%	99.5%	99.5%
avgIoU—Bow	89.8%	95.1%	84.9%	88.5%	86.9%	85.8%	93.9%	81.3%	86.5%	90.4%
avgIoU—RH	92.5%	76.9%	76.6%	74.3%	82.6%	87.2%	81.7%	74.8%	90.4%	67.8%
F1 score	1.00	1.00	1.00	1.00	0.97	1.00	0.98	0.97	1.00	1.00
Recall	1.00	1.00	1.00	1.00	1.00	1.00	1.00	0.95	1.00	1.00
Total frames	812	447	1941	1947	827	841	866	1462	3145	1499
Bow detected	812	447	1941	1935	803	791	866	1440	3131	1499
	(100%)	(100%)	(100%)	(99%)	(97%)	(94%)	(100%)	(98%)	(99%)	(100%)
RH detected	807	447	1931	1931	788	841	866	1439	3140	1456
	(99%)	(100%)	(99%)	(99%)	(95%)	(100%)	(100%)	(98%)	(99%)	(97%)
Both detected	626	447	1931	1919	764	791	866	1429	3126	1456
	(99%)	(100%)	(99%)	(98%)	(92%)	(94%)	(100%)	(97%)	(99%)	(97%)

**Table 6 sensors-25-00674-t006:** Performance metrics obtained by the study participants.

Group	Learner	Assist.	BLS				BSB		
				Pull	Push	Inter	Union	IOU	${\mathrm{IO(U)}}^{2}$
G1	A	Yes	90.7%	0.32	0.27	0.27	0.32	84%	2.64
		No	72.5%	0.21	0.19	0.18	0.22	82%	3.72
	B	Yes	98.1%	0.20	0.17	0.17	0.20	85%	4.25
		No	88.9%	0.16	0.18	0.16	0.18	89%	4.94
	C	Yes	92.3%	0.37	0.32	0.32	0.37	86%	2.34
		No	72.7%	0.50	0.47	0.47	0.50	94%	1.88
G2	D	No	34.7%	0.51	0.52	0.50	0.53	94%	1.78
		Yes	96.4%	0.28	0.26	0.26	0.28	93%	3.32
	E	No	35.1%	0.28	0.24	0.24	0.28	86%	3.06
		Yes	86.9%	0.27	0.27	0.27	0.27	100%	3.70
	F	No	77.8%	0.36	0.31	0.30	0.37	81%	2.19
		Yes	89.9%	0.26	0.23	0.22	0.27	81%	3.02

## Data Availability

Data are contained within the article.

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
