# Peer review of "A Posture Training System Based on Therblig Analysis and YOLO Model—Taking Erhu Bowing as an Example"

_sensors, 2025, doi:10.3390/s25030674_

Round 1
Reviewer 1 Report (New Reviewer)
Comments and Suggestions for Authors
I commend the authors for this work. However, this needs to be improved. Please see detailed comments below.
1. The model mainly relies on front-facing video data. While effective in this context, it may face challenges with non-standard camera angles or different environments. This limitation should be discussed in detail.
2. While the Therblig analysis is technically sound, its contribution can be seen as overly complex for this application without comparative evidence. This need more justification.
3. While the system achieves high detection rates there were no details are about the variability in environmental factors or if the preprocessing will address these type of changes.
4. If available I recommend authors to provide a baseline comparison with alternative posture-training systems or simpler object-detection methods to substantiate the value of the YOLO-Therblig framework.
5. Simplify and clarify the role of Therblig analysis for readers that may not be familiar with this concept. A concise explanation of its benefits over other motion analysis techniques would help.
6. Acknowledge the controlled nature of the testing environment as a limitation. If you have any plans to expand dataset, discuss them and inform if they address generalizability issues
7. Highlight how this system could be scaled or adapted to other musical instruments or similar posture-training tasks.
Comments on the Quality of English Language1. Certain sentences are unnecessarily long or repetitive.
2. Transitions between sections, particularly between results and discussion, could be smoother to enhance the narrative flow.
Author Response
Please see attached file.

Reviewer 2 Report (New Reviewer)
Comments and Suggestions for Authors
The manuscript proposes an evaluation method for erhu performance based on Therblig Analysis and YOLO, which is somewhat innovative in its application. However, the study still has the following deficiencies.
1. YOLO, as a primary method for object detection, is more commonly applied to qualitative analysis, i.e., recognizing whether an action is present or absent. However, the recognition of posture and action standards is highly abstract, and the standardization of actions is more about quantitative analysis, i.e., assessing the degree to which key actions are performed correctly.
2. The evaluation of erhu performance primarily focuses on the fluidity of a series of actions, which constitutes a time-series analysis. The manuscript decomposes erhu performance into multiple static postures and scores them based on the proportion of these postures. While this approach can evaluate the standardization of erhu performance to a certain extent, it is more suitable for evaluating beginner-level performances. For erhu players with some foundation, this type of evaluation holds limited significance, as they do not have any missing basic posture or obvious posture errors.
3. The overall innovation of the paper still appears to be lacking, as the author did not make any improvements to the YOLO algorithm, and the proposed Scoring Methods are relatively simple. I suggest that the author make targeted improvements to the YOLO method based on the characteristic that the standardization of erhu performance gestures may be reflected in subtle differences in posture. By doing so, the algorithm can better distinguish between similar actions with subtle differences, transforming the quantitative recognition problem of posture standardization into a classification problem, which has more practical significance.
Author Response
Please see attached file.

Reviewer 3 Report (New Reviewer)
Comments and Suggestions for Authors
This paper has designed a posture training system based on therblig analysis and YOLO for Erhu bowing learning. The topic is interesting. The methods for applying YOLO and therblig analysis are detailed and meaningful.
The reviewer has the following comments.
1. The technological novelty is limited or vague. It is difficult to identify the novelty of the YOLO-OD model from the existing YOLO models, particularly from a network architecture point of view. In addition, in Section 5, the proposed and existing algorithms should be pointed out clearly.
2. More recent references should be included.
Round 2
Reviewer 2 Report (New Reviewer)
Comments and Suggestions for Authors
The revised version of the manuscript provides explanations for some issues. It is suggested that the authors collect more videos of erhu performers in subsequent research to demonstrate the effectiveness of the method and identify any problems that may exist in its application.
This manuscript is a resubmission of an earlier submission. The following is a list of the peer review reports and author responses from that submission.
Round 1
Reviewer 1 Report
Comments and Suggestions for Authors
this study propose a bow tracking model, and developed a training system with warnings for the novel erhu players. A experimental process is designed and YOLO model is used to capture and predict bow movements and positioning.
I did not understand very well about the relations of Therblig analysis and the YOLO based position detection process. It is really necessary to incorporate Therblig ?
Author Response
|
Response to Reviewer 1 Comments
|
||
|
1. Summary |
|
|
|
Thank you very much for taking the time to review this manuscript. Please find the detailed responses below and the corresponding revisions/corrections highlighted changes in the re-submitted files.
|
||
|
2. Questions for General Evaluation |
Reviewer’s Evaluation |
Response and Revisions |
|
Does the introduction provide sufficient background and include all relevant references? |
Yes |
|
|
Are all the cited references relevant to the research? |
Yes |
|
|
Is the research design appropriate? |
Can be improved |
|
|
Are the methods adequately described? |
Yes |
|
|
Are the results clearly presented? |
Can be improved |
|
|
Are the conclusions supported by the results? |
Yes |
|
|
3. Point-by-point response to Comments and Suggestions for Authors |
||
|
Comments 1: this study propose a bow tracking model, and developed a training system with warnings for the novel erhu players. A experimental process is designed and YOLO model is used to capture and predict bow movements and positioning. I did not understand very well about the relations of Therblig analysis and the YOLO based position detection process. It is really necessary to incorporate Therblig ?
|
||
|
Response 1: Thank you for pointing this out. We have added the following explanation for using Therblig analysis in this research into Section 1 (introduction) - page 2 Line 68 - 78.
“When designing a training system related to motion recognition, it is necessary to decompose the actions to thoroughly explore the sequence of movements and the relevant objects that need to be detected. Industrial engineering provides various methods for motion decomposition, such as visual motion analysis, operation procedure analysis, Therblig analysis, film analysis, and image analysis. Among these, Therblig analysis is most commonly used to break down actions into basic elements, where complex body movements can be viewed as combinations of various motion elements. Therblig analysis also highlights the objects manipulated by the operator in each action. Therefore, when identifying images depicting mechanical motion, conducting a Therblig analysis beforehand to break down the machinery elements of movement and identify related components will aid in the systematic identification and assessment of this mechanical action.”
|
||
|
4. Response to Comments on the Quality of English Language Comments: English language fine. No issues detected.
5. Additional clarifications |
||
|
(1) Since the practice of body movements or related technical movements traditionally requires visual observation by instructors before giving advice, this limits the location or time period for operators to practice. (2) Therblig elements define the fundamental components of human movement. Therblig analysis also highlights the objects manipulated by the operator in each action. Complex body movements can be seen as combinations of various Therbligs. Therefore, when identifying images depicting mechanical motion, conducting a Therblig analysis beforehand to break down the machinery elements of movement and identify related components will aid in the systematic identification and assessment of this mechanical action. (3) The objective of this research is to utilize image recognition technology for the identification of mechanical movements, and to provide scoring and feedback regarding the accuracy of these movements. Consequently, this paper introduces an architecture that incorporates Therblig analysis to enhance the recognition of mechanical movements. |
||

Reviewer 2 Report
Comments and Suggestions for Authors
The author proposes an educational support system using yolo. However, it was not possible to determine from the results whether this system was effective. I think the explanation of the results needs to be fundamentally reconsidered.
1. Adopts unique evaluation indicators such as BLS, BSS, and BSB. Where are these metrics used in the results?
2. I don't understand how to read Figure 10.
3. The limitations of the discussion section were not clear. (Bullets 1. to 5.)
4. It was not clear how to derive the results in the discussion section.
Author Response
|
Response to Reviewer 2 Comments
|
||
|
1. Summary |
|
|
|
Thank you very much for taking the time to review this manuscript. Please find the detailed responses below and the corresponding revisions/corrections highlighted changes in the re-submitted files.
|
||
|
2. Questions for General Evaluation |
Reviewer’s Evaluation |
Response and Revisions |
|
Does the introduction provide sufficient background and include all relevant references? |
Yes |
|
|
Are all the cited references relevant to the research? |
Yes |
|
|
Is the research design appropriate? |
Yes |
|
|
Are the methods adequately described? |
Yes |
|
|
Are the results clearly presented? |
Must be improved |
|
|
Are the conclusions supported by the results? |
Must be improved |
|
|
3. Point-by-point response to Comments and Suggestions for Authors |
||
|
Comments 1: The author proposes an educational support system using yolo. However, it was not possible to determine from the results whether this system was effective. I think the explanation of the results needs to be fundamentally reconsidered.
|
||
|
Response 1: Thank you for pointing this out. In response, we have revised the description of the performance indicators in Section 6. More detailed explanations on how to use the BLS, BSS, and BSB indicators have been incorporated into Subsection 6.2 (Scoring Methods), specifically on page 13 (lines 378–388, 397–410, 414–420) and page 14 (lines 421–436).
|
||
|
Comments 2: Adopts unique evaluation indicators such as BLS, BSS, and BSB. Where are these metrics used in the results? |
||
|
|
||
|
Response 2: More detailed explanations on how to use the BLS, BSS, and BSB indicators have been incorporated into Subsection 6.2 (Scoring Methods), specifically on page 13 (lines 378–388, 397–410, 414–420) and page 14 (lines 421–436). |
||
|
Comments 3: I don't understand how to read Figure 10.
|
||
Response 3:
Figure 10 visualizes BSB, providing instructors and learners with the ability to analyze erhu bowing issues and propose appropriate improvement plans. Key information presented in the image includes:
- Trajectory of bow slope during bow movement
- Offset position of BSB relative to the baseline
- BSB overlap range between bow pull and bow Push
More detailed explanations on BSB chart have been incorporated into Subsection 6.2 (Scoring Methods), specifically on page 13 (lines 414–420) and page 14 (lines 421–436).
|
Comments 4: The limitations of the discussion section were not clear. (Bullets 1. to 5.)
|
Response 4:
Thank you for pointing this out. We have reevaluated and divided the potential future studies into two parts: addressing two limitations that need to be overcome and exploring the inclusion of three additional functions. These recommendations are detailed specifically on page 15 (lines 450 – 484).
|
Comments 5: It was not clear how to derive the results in the discussion section.
|
Response 5:
Thank you for pointing this out. we have revised the description of Section 7 to make the wording clearer to support our results and future work, specifically on page 14 (lines 438–447) and page 15 (lines 448–489).

Reviewer 3 Report
Comments and Suggestions for Authors
In this paper, a computerized system based on the YOLO-OD model was developed for erhu bowing training. First, Therblig analysis was performed to segment the erhu bowing process into multiple steps. Second, the YOLO-OD model was developed to detect and track the critical objects identified in video frames. Third, scoring methodologies were developed for bow level and bow straightness. The research is well thought out and worked on. But I have some suggestions.
1)The author should point out the problems with existing methods, rather than simply listing existing methods in the Section 1.
2)In the experimental results, the author should provide more comparison results with the state-of-the-art methods.
3)The author gave examples of straightness analysis report for erhu bowing, but not providing sufficient analysis and explanation for the results
4)Reviewing throughout the paper, the English grammar, spelling, and sentence structure still needs improve.
Moderate editing of English language required.
Author Response
|
1. Summary |
|
|
|
Thank you very much for taking the time to review this manuscript. Please find the detailed responses below and the corresponding revisions/corrections highlighted changes in the re-submitted files.
|
||
|
2. Questions for General Evaluation |
Reviewer’s Evaluation |
Response and Revisions |
|
Does the introduction provide sufficient background and include all relevant references? |
Can be improved |
|
|
Are all the cited references relevant to the research? |
Can be improved |
|
|
Is the research design appropriate? |
Can be improved |
|
|
Are the methods adequately described? |
Can be improved |
|
|
Are the results clearly presented? |
Can be improved |
|
|
Are the conclusions supported by the results? |
Can be improved |
|
|
3. Point-by-point response to Comments and Suggestions for Authors |
||
|
Comments 1: In this paper, a computerized system based on the YOLO-OD model was developed for erhu bowing training. First, Therblig analysis was performed to segment the erhu bowing process into multiple steps. Second, the YOLO-OD model was developed to detect and track the critical objects identified in video frames. Third, scoring methodologies were developed for bow level and bow straightness. The research is well thought out and worked on. But I have some suggestions. 1)The author should point out the problems with existing methods, rather than simply listing existing methods in the Section 1.
|
||
|
Response 1: Thank you for pointing this out. We have added two paragraphs to explain the problems of existing learning methods into Section 1 (introduction), specifically on page 1 (lines 18 – 23) and page 2 (lines 59 – 64).
|
||
|
Comments 2 2)In the experimental results, the author should provide more comparison results with the state-of-the-art methods. |
||
|
|
||
|
Response 2 Thank you for highlighting this issue. We have expanded Table 2 to include additional YOLO models as well as RTDETR and have explained that all models achieved perfect performance. Consequently, we opted for the more efficient option, YOLOv8n, due to its lighter computational requirements. We also revised and added explanation of the model training results, specifically on page 7 (lines 255 – 257) and page 8 (lines 258, 261 – 262, 265 - 278).
|
||
|
Comments 3 3)The author gave examples of straightness analysis report for erhu bowing, but not providing sufficient analysis and explanation for the results |
||
Response 3
Thank you for pointing this out. In response, we have revised the description of the performance indicators in Section 6. More detailed explanations on how to use the BLS, BSS, and BSB indicators have been incorporated into Subsection 6.2 (Scoring Methods), specifically on page 13 (lines 378–388, 397–410, 414–420) and page 14 (lines 421–436).
|
Comments 4 4)Reviewing throughout the paper, the English grammar, spelling, and sentence structure still needs improve.
|
Response 4
Firstly, I apologize for any lack of fluency in my English, as it is not my first language. Prior to submission, I sought assistance from Wallace Academic Editing (https://www.editing.tw/) for English editing. Additionally, I have used Google Translate and textGPT for grammar and spelling checks. If there are any further corrections or improvements needed in my wording, I am open to making revisions based on your recommendations.
- Response to Comments on the Quality of English Language
Comments: Moderate editing of English language required.
Response: I sought assistance from Wallace Academic Editing (https://www.editing.tw/) for English editing. Additionally, I have used Google Translate and textGPT for grammar and spelling checks. If there are any further corrections or improvements needed in my wording, I am open to making revisions based on your recommendations.

Round 2
Reviewer 2 Report
Comments and Suggestions for Authors
I have reviewed the answer. However, since we currently lack the results necessary to assess this paper, we cannot determine whether it is good or bad.
The issue arises from the fact that YOLO8N is an existing method, and there is no novelty in its approach. Nevertheless, I have come across information suggesting that its uniqueness and effectiveness stem from its application in teaching stringed instruments.
If that is the case, Figure 10 should serve as an evaluation of the proposed method, but what the paper describes is merely a sample display (presumably showcasing the most impressive results). I am apprehensive about the potential for cherry-picking. To demonstrate the novelty and effectiveness of applying this method to stringed instruments, we believe it is essential to evaluate the disparity between correct data and predicted data using a general index such as the root mean squared error.
Furthermore, despite the claim that it is effective for education, this assertion is solely based on the author's description, without any actual evaluation of its educational impact.
In summary, the content of this paper appears disconnected from its intended claims, and its evaluation seems biased. Consequently, a fair assessment of this paper is not possible at this time.
Reviewer 3 Report
Comments and Suggestions for Authors The authors have addressed my concerns, so I suggest accepting it as a regular paper.